# Plasmonic Metal Nanoparticles in Sensing Applications: From Synthesis to Implementations in Biochemical and Medical Diagnostics

**DOI:** 10.3390/molecules30244745

**Published:** 2025-12-12

**Authors:** Grace Nemeth, Jacob Speers, Salman Shaheen, Vladimir Kitaev

**Affiliations:** Department of Chemistry and Biochemistry, Wilfrid Laurier University, 75 University Ave. West, Waterloo, ON N2L 3C5, Canada

**Keywords:** plasmonic metal nanoparticles, localized surface plasmon resonance, plasmonic sensing substrates, colorimetric sensing, synthesis and transformations of anisotropic metal morphologies

## Abstract

This work overviews recent (last 3–4 years) advances in sensing based on localized surface plasmon resonance (LSPR) of plasmonic metal nanoparticles (PMNPs). Starting with a brief background, recent reviews in the field and relevant related areas are summarized. Next, recent progress in PMNP synthesis and post-synthetic transformations is discussed in the context of PMNP sensing performance. Subsequently, preparation of sensing substrates based on PMNPs is examined. Recent developments in colorimetric and LSPR sensing constitute the core of the review material with the focus on implementation of PMNPs and their sensing modalities. Advances in other sensing methods with direct relevance to PMNP implementations are also highlighted in the context. Perspectives on directions of further advances in LSPR sensing with PMNPs and overcoming existing limitations conclude this review.

## 1. Introduction

### 1.1. Brief Background on PMNP Properties and Applications

Out of the great diversity of nanomaterials with their unique size-dependent properties, plasmonic metal nanoparticles, PMNPs, stand out due to prominent optical characteristics that enable their impressive range of applications [1,2]. Free or delocalized electrons of metals are essential for PMNPs, as well as chemical stability to resist PMNP degradation. Silver and gold are two metals that meet these two criteria and thus are dominant materials of PMNPs. Upon nanoscale confinement, electron oscillations (localized surface plasmon resonance, LSPR) become size- and shape-dependent [3]. This nanoscale size dependence of strong light interactions with PMNPs not only can be esthetically appealing (Figure 1) but it can also serve as one of the quantum size dependences that is simple and natural to visualize and to intuitively understand. Not surprisingly, considering the use of noble metals starting at the dawn of humanity, PMNPs are one of the earliest nanomaterials documented in the creation of a unique piece of art, a Lycurgus cup of the 4th century (Figure 1A–C) [4,5]. LSPR of PMNPs are responsible for the distinctive dichroic appearance of the Lycurgus cup (Figure 1A,B) and similar strong scattering PMNP dispersions (Figure 1D,E), when viewed in reflected and transmitted light.

With respect to the applications of PMNPs, size- and shape-dependence of LSPR enables convenient LSPR tuning and maneuverability during PMNP synthesis and post-synthetic transformations. Here, advantages of anisotropic PMNPs over isotropic ones (that extend uniformly in all three orthogonal directions and hence are classified as 3-D shape/morphologies), such as spherical, cubic, octahedral, etc., can be briefly pointed out. More effective tuning of LSPR energy/wavelength can be achieved via size variation in just a single dimension, such as the length for 1-D nanorods, compared to 3-D PMNPs (see Figure 2 and Section 3 for more details). Consequently, realization of the size and shape control in synthesis and post-synthetic transformations of PMNPs is crucial for efficient practical implementations of PMNPs in sensors. Given that *d-sp* transitions of silver and gold are in near UV and visible, respectively, most PMNPs have LSPR in the visible and near IR range, so PMNPs are conveniently suitable for colorimetric sensing. The ability to trace LSPR maxima with the spectral resolution at the level of 0.01–0.02 nm brings powerful capabilities to PMNP-based LSPR sensing [10] which is appreciably technically simpler compared to conventional SPR instrumentation [11]. Surface-enhanced Raman spectroscopy (SERS) takes advantage of the strong electromagnetic field enhancement in the vicinity of PMNPs [12] and, especially, their assemblies to enable signal enhancement on the order of 10 [10] and thus single-molecule detection using Raman spectroscopy [13]. Other diverse applications of PMNPs include catalysis [14,15], photonics (enhanced emission [16] and adsorption, photoelectric sensing [17], photovoltaics [18], etc.) and biomedicine [19] (diagnostics and therapy).

### 1.2. Overview of the Recent Progress in the Field and Scope of This Review

Prior to defining the scope of our review, we briefly summarize recent developments and the state of the art in the reviews of this field for the last ca. three years. General progress in LSPR developments has been comprehensively summarized by Mcoyi et al. [2]. A very good background review on LSPR sensing with emphasis on soft matter has been presented by Zhdanov [20]. Plasmonic sensing with silver NPs has been thoroughly discussed by Li et al. [21]. In their recent work, Lin et al. presented a comprehensive overview of recent advances in LSPR sensor technologies [22]. Another detailed review on plasmonic biosensors by Hamza et al. was published in 2022 [23]. Comparison of SPR and LSPR sensing and advances in real time detection are reviewed by Cho et al. [11]. Sensing based on morphological changes in metal is comprehensively overviewed by Zhang et al. [24]. Opportunities and challenges of plasmonic sensors have been thoroughly discussed by Das et al. [25]. OpenSPR sensing based on PNMPs has been published by Hanson et al. [10]. Plasmonic sensing with the emphasis on pharmaceutical and biomedical applications was recently overviewed by Akgonüllü and Denizli [26]. An excellent review on applications of PMNPs in point-of-care diagnostics has been made by Geng et al. [27]. For reviews in related and overlapping fields with PMNP-based sensing, Wu et al. [28] discussed colorimetric sensing, where plasmonic nanostructures are an important part. Hydrogel sensing, with an extended part on PMNPs, has been reviewed by Song et al. [29]. The interplay of PMNPs and fluidics was comprehensively discussed by Bhalla et al. [30]. Dark field sensing with plasmonic nanomaterials has been nicely covered by Zhang et al. [31] Kant et al. [32] delivered a comprehensive review that discussed the recent progress of PMNP-based sensors, challenges faced in the field and future outlook. Finally, a broader perspective on plasmonic nanostructures that include not only PMNPs but other nanomaterials, such as graphene, has been presented by Wu et al. [33]. Appendix A in Appendix A summarizes the recent reviews described above.

In this review, with the goal to complement current reviews and to expand upon them in several areas, we focus on pathways from PMNP preparation to sensor fabrication and then to colorimetric and LSPR sensing based on PMNPs. One of the points emphasized in this review is PMNP implementation in sensing technologies. In this context, in addition to LSPR sensing, some SERS examples are discussed when relevant in the context. For complementary reviews, an excellent recent in-depth perspective on SERS sensing [9] and recent reviews on fabricated metal nanostructures for LSPR sensing [33], PMNPs in fibre optic sensor geometries [11] and LSPR applications of PMNPs in chemical and biological sensing using fibre optics [34] can be recommended.

## 2. PMNP Synthesis and Post-Synthetic Transformations Relevant to Sensing

Since the early 2000s, the field of PMNP synthesis has progressed significantly, with major advances in the mechanistic understanding and control of nanoparticle size and morphology. Fundamental reviews by Rycenga et al. [35] and Regura et al. [36] provide a comprehensive reference of silver nanoparticle syntheses, systematizing the broad range of attainable morphologies of silver PMNPs. In comparison, gold NPs have been investigated extensively for biological and biomedical applications, but their accessible shape diversity is comparatively limited, as well as the ease of synthesis, as reflected in reviews, e.g., by Dykman and Khlebtsov [37]. Diverse applications of gold NPs have been discussed in the recent review by Karnwal et al. [38].

### 2.1. Anisotropic PMNP Shapes

For the present progress in PMNP synthesis, Figure 3 showcases several seed-mediated strategies for producing advanced PMNPs with tunable optical properties advantageous for LSPR sensing. LSPR tunability is more easily accessible for anisotropic PMNPs with a shape and size selection. High-purity gold nanotriangles [39] (AuNTs) (Figure 3A) serve as a good example of such 2-D anisotropic PMNPs. Key features of AuNT synthesis are that by controlling reaction kinetics through high concentrations of absorbing species: cetyltrimethylammonium chloride (CTAC) and KI, tunable sizes and uniformity can be attained. Similarly, Carone et al. [40] used cetyltrimethylammonium bromide (CTAB) to direct growth of AuNPs into several different shapes, including quasi-spherical, rod-like, and pentagonal bipyramids. Another notable feature of this work is AuNT purification through self-assembled microfibers that separates impurity shapes in an entropically driven process, thus improving the resulting shape and size uniformity [39]. Similarly, Podlesnaia et al. [41] reported microfluidics-based synthesis of AuNTs that delivers precise seed control, efficient two-step growth and scalable production with improved uniformity. In another example that combines 1-D structures and shell formation, hollow AuAg nanorods were fabricated by galvanic exchange using silver nanorods as a template (Figure 3B) [42]. The prepared PMNPs demonstrated high chemical stability due to gold-rich outer regions and silver near internal cavities, resulting in enhanced SERS activity. In a third example of more complex anisotropic morphologies, AuAg nanorattles (Figure 3C) were synthesized to deliver LSPR spanning from 1000 to 3000 nm, thus fully covering the near IR region [43]. Synthetically, AuAg nanorattles were formed by combining galvanic replacement and chemical reduction in a seed mixture of gold bipyramids and Au/Ag penta-twinned nanorods by using ascorbic acid as a reducing agent and tuning the rate of HAuCl_4_ addition [43]. Electron microscopy, spectroscopy and modelling of these PMNPs revealed strong IR signal enhancement, showcasing their potential for LSPR sensing, as well as optical communication [43]. As a more intricate example of shape selection, gold nanorods with five-fold rotational symmetry were prepared (Figure 3D,E), where end-to-end dimers retained strong chiro-optical signals advantageous for chiral LSPR sensing [44]. An example of a scalable synthesis of chiral PMNPs with an excellent size and shape selection employed chiral amino acids and peptides to yield 432-symmetric helicoid PMNPs (Figure 3F) with a high g-factor of 0.2, suitable for polarization control, chiral sensing and thus a broad range of biomedical applications [45,46] and other applications, e.g., hydrogen sensing [47]. Zhao et al. developed Au nanoheptamers composed of six Au nanospheres interconnected by thin metal bridges around a central core, forming strong electromagnetic hot spots for enhanced near-field effects [48]. Vinnacombe-Willson et al. [49] prepared gold nanostars (AuNSts) that exhibited high extinction in the near-infrared (NIR) region with advantages in sensing due to the electromagnetic fields at their tips and their high photothermal performance. AuNSts were then embedded into a hydrogel matrix that has paved the way for the development of unique biosensors, drug delivery systems and actuators [49].

### 2.2. Shells and Hollow PMNP Morphologies

In addition to anisotropic structures, shells and related hollow nanostructures offer additional dimension in LSPR engineering and resulting sensing capabilities, e.g., as it has been already shown in discussion of Figure 3B. Recent progress in plasmonic nanostructure engineering highlights how careful control over shell composition, thickness and interfacial chemistry can generate multifunctional sensing platforms with finely tunable LSPR. Light-assisted colloidal growth using silica-encapsulated gold bipyramids as localized photothermal sources has enabled the low-temperature synthesis of anisotropic iron oxide, silver and palladium NPs, illustrating the versatility of plasmonic heating for solution-phase growth of complex nanostructures [50]. Core–shell PMNPs that incorporate Raman reporters—such as silver core–silica shell NPs with 4-mercaptobenzonitrile in the interfacial layer—enable interference-free SERS detection of membrane type-1 matrix metalloproteinase (MT1-MMP) in breast cancer cells, overcoming spectral overlap and enhancing specificity for clinical diagnostics [51]. Similarly, AgNPs with ultrathin gold shells embedded with 4-mercaptobenzoic acid (AgMBA@Au) have been integrated into lateral-flow immunoassays for colorimetric and SERS dual-mode detection of SARS-CoV-2 IgG, achieving picogram-per-millilitre sensitivity and outperforming traditional ELISA and colloidal-gold strips in clinical samples [52]. Beyond linear optical responses, chiral gold nanorods and their silver or titania shell derivatives exhibit tunable plasmonic circular dichroism bands spanning the UV–NIR region [53]. These studies elucidate chirality transfer mechanisms and enable the rational design of chiral-optically active plasmonic nanomaterials for circularly polarized light applications [53]. Bioinspired, peptide-mediated one-pot syntheses offer another route to multifunctionality: a single short peptide can reduce and stabilize gold, silver and Au@Ag NPs under ambient conditions to yield shell morphologies while also conferring selective, pH-dependent responses to Hg^2+^, Fe^2+^ and Mn^2+^ ions [54]. At the assay level, etchable silver shells on gold nanobipyramids support a plasmonic ELISA with multi-colorimetric readout for C-reactive protein (CRP), where the silver shell thickness directly controls the sensitivity of LSPR peak shifts and enables sub-nanogram detection both spectroscopically and by naked eye detection [55]. Extending these principles to larger architectures, hydrophobic microcapsules consisting of a liquid core surrounded by an inert nanometre-thin silica shell embedding mixtures of Ag and Au NPs exhibit mechanically robust, optically tunable systems suitable for SERS studies, photothermal therapy, interfacial electrocatalysis, antimicrobial coatings and drug delivery since their optical response can be precisely tuned by the Ag/Au ratio and shell thickness [56].

### 2.3. PMNPs with Magnetic Capabilities

Magnetic functionality integrated with PMNPs offers a powerful feature of moving and separating resulting nanocomposites by an external magnetic field that is highly advantageous for multiple sensing assays. To avoid LSPR quenching with strongly light-absorbing magnetic materials, plasmonic-magnetic nanocomposites need to be carefully designed. Of the recent work, star-like Au@Fe_3_O_4_ core–shell nanostructures provide a highly uniform magnetite shell with superior saturation magnetization, a large red shift in the LSPR into the first biological window and demonstrable magneto-mechanical and photothermal effects in cell cultures [57]. Theoretical work further suggests that Fe_3_O_4_@M (M = Ag or Au) core–shell NPs coupled to metal films could serve as tunable plasmonic nanolasers, with their emission wavelength controllable by NP size, gap distance and an external magnetic field [58]. Michalowska and Kudelski [59] recently deposited a thin layer of silica (3 nm) to the surface of plasmonic-magnetic nanostructures (Fe_3_O_4_@Ag)@SiO_2_) that exhibited lower deviation of the mean values when applied to SHINERS (shell-isolated nanoparticle-enhanced Raman spectroscopy). Together, these studies highlight the versatility of magnetic-plasmonic nanomaterials as multifunctional platforms for sensing, imaging, therapy and tunable photonic devices.

## 3. Structure–Property Relationships for Sensing Applications of PMNPs

To start, we expand upon advantages of anisotropic PMNPs from the introduction and several previous sections. Scientific studies of PMNPs, as many other NPs, historically started with “spherical” particles. Yet, unlike true spherical shapes of amorphous or polycrystalline materials, e.g., of latex or silica, metal NPs are not truly spherical due to underlying crystal lattices of metals, e.g., face-centred cubic for silver and gold. “Spherical” description has also been reenforced by resolution limitations of electron microscopy in earlier studies. High-resolution images of gold and silver NPs show multifaceted surfaces of different crystallographic planes [60,61]. These different confining planes may complicate chemical modification and catalytic applications. Thus, “quasispherical” would be a more fitting description of these NPs. The two main most “spherical” single-crystal morphologies known for metals are icosahedral (I_h_) and cuboctahedral (O_h_). Correspondingly, to obtain truly uniform “spherical” PMNPs, a single nuclei type is essential. Any twinned defects, common in PMNPs, translate into the shape non-uniformities that are apparent in high-resolution EM images of “spherical” PMNPs. So, effectively, true shape and size selection of “spherical” PMNPs is similarly as demanding as for other morphologies, only the rounding and EM resolution mask some of the imperfections.

Next, for the efficiency of size variation, which is essential for LSPR tuning, the scaling laws are instructive to compare. For the growth of isotropic (3-D) NPs taking place uniformly in all three orthogonal dimensions, NP size increases proportionally to the cubic root of the amount of material used for the size increase (Figure 4A). For 2-D morphologies growing in two orthogonal directions of a plane, the size increases scales as a square root of the amount of material (Figure 4B). Correspondingly, for 1-D morphologies of nanorods elongating in a single dimension, there is a direct proportionality between the size increase and the amount of material used (Figure 4C). Thus, 1-D PMNPs are most effective for size (and LSPR) tuning followed by thin 2-D ones. To avoid possible confusion, it can be noted that “3-D”, “2-D” and “1-D” are explicitly referring to the intrinsic properties of NPs as objects [62], while relative to (considered to be infinite) bulk materials, NPs can be described as 0-D objects with resulting “quantum” nanoscale properties [63].

Upon PMNP size increase, the LSPR, as a surface phenomenon, increases in peak wavelength (“red shift”), decreasing in energy, as schematically shown in Figure 4. Comparing changes in LSPR with changes in PMNP dimensions, LSPR shifts are smaller for spherical PMNPs compared to anisotropic PMNPs since plasmon energy is spread in all three dimensions (Figure 4A). Correspondingly, LSPR changes with size are the strongest for length changes in 1-D rods (Figure 4C), while 2-D PMNPs have intermediate values between spheres and rods (Figure 4B). Absorption cross-sections of PMNPs are dependent on their dimensions. Generally, as length/width/size increases, the scattering and absorption cross-section also increases [64]. Rational design of PMNP shapes for specific applications [65] is an important area that will see more growth upon AI utilization.

Sensing with PMNPs is based on LSPR response to changes in the dielectric media at PMNP interfaces. As the refractive index of PMNPs’ surroundings is changed, the LSPR shifts, increasing in wavelength with the increase in refractive index, as shown in Figure 5. The magnitude of the LSPR change per refractive index change (nm/RIU) is defined as sensitivity. Sensitivity is an inherent property of PMNPs and is an important parameter in plasmonic sensing, particularity in sensor design and optimization. Similarly to the LSPR changes upon size increase, the sensitivity is highest for 1-D PMNPs with LSPR localized in a single dimension, next for 2-D PMNPs, and the lowest for 3-D spherical PMNPs. Typical values for the latter are 150–200 nm/RIU [66], while for nanorods, values as high as 500 nm/RIU were reported [10,67].

## 4. Fabrication of Sensing Substrates and Sensors Using PMNPs

Most sensing with PMNPs is performed using solid surfaces rather than dispersions, so PMNP deposition on a substrate/surface is an essential part of sensor fabrication. PMNP monolayers are most common, unless porous separating media are used. Three different scenarios can be schematically outlined for monolayers, as shown in Figure 6. First, when PMNPs are clustered together (Figure 6A), individual LSPR properties are appreciably lost due to interparticle interactions, and the resulting sensing properties are poor. In a most practically common scenario, PMNPs are deposited on a substrate in such a way that individual particles are appreciably separated, while their surface distribution is random (Figure 6B). The individual PMNP properties are largely preserved which allows for good sensing properties. Yet, the ideal scenario is when PMNP interparticle distances are tightly controlled (as well as PMNP dimensions and orientation)—such substrates (Figure 6C) enable strong collective LSPR modes with the postulated narrow resonances [68]. The latter scenario is still less common for PMNPs compared to fabricated/top-down nanostructures.

Of the many ways to prepare LSPR sensors, Sun et al. [69] described an effective approach for fabricating sensing substrates less affected by interparticle separation by using hollow gold nanoshells. To fabricate the sensor (Figure 7A), a cleaned glass slide was soaked in APTES (3-aminopropyltriethoxysilane), and hollow AuNPs were deposited to bind to a thiolated surface [69]. The sensitivity of the resulting substrates was reported to be 360 nm/RIU, which is appreciably higher compared to solid AuNP sensors prepared by the same method [69]. Thus, it has been shown that hollow shells are much less sensitive to interparticle separation and can serve as promising morphologies for LSPR sensing, with benefits of preventing agglomeration, ease of functionalization, sensor regeneration and possibilities for integration with various systems including microfluidics [69]. Ma et al. [70] reported on an integrated microfluidic LSPR chip, with a schematic of the fabrication process highlighting the sequential steps from nanomaterial preparation to device assembly shown in Figure 7B. First, a thin gold layer (8 nm) is deposited onto a cleaned quartz substrate via thermal evaporation, followed by thermal annealing at 560 °C for 6 h to induce dewetting and form AuNPs with uniform morphology [70]. Concurrently, polydimethylsiloxane (PDMS) is cast into a mould, cured and demoulded to produce microfluidic channel structures [70]. These PDMS microchannels are then bonded to the nanoparticle-patterned quartz substrate, forming sealed microcavities above the AuNP arrays that serve as localized plasmonic sensing regions [70]. Process pipes are incorporated into the PDMS to allow controlled liquid delivery through the sensing chambers, enabling high throughput biosensing [70]. The integration of microfluidics with LSPR-active NP arrays enables precise control of sample flow, reduced reagent consumption and multiplexed detection within a compact chip [70]. This fabrication approach delivers a robust platform for label-free, sensitive biosensing by coupling the nanoplasmonic properties of AuNPs with the practical functionality of microfluidic systems [70]. Another recent work describing fabrication of sensing substrates and sensors using PMNPs is by Jeong et al. [71] who prepared electrodes combining AgNPs with laser induced graphene (AgNP/LIG) for multi-detection of select heavy metal ions.

## 5. Colorimetric Sensing with PMNPs

Colorimetric sensing methods are now well established in point-of-care detection, environmental monitoring and disease control [72,73,74]. This sensing platform provides the advantages of fast response, ease of use, detection by naked eye and low cost [72,73,75]. Colorimetric sensing with PMNPs offers advantages of well-defined colours and strong colour changes by virtue of the LSPR [65]. Colorimetric sensing with integrated PMNPs relies upon interactions with target analytes or changes in the external environment leading to aggregation or de-aggregation in dispersions, alterations of morphology, and surface composition of PMNPs, leading to shifts in LSPR peaks, shown schematically in Figure 8, that manifest as intense visible colour changes [65,72,76]. AuNPs are often used for colorimetric sensing due to ease of surface functionalization and chemical stability [77,78]. There are several subcategories of PMNP-based colorimetric sensing mechanisms including PMNP growth [79], aggregation [72], surface modification/functionalization [78] and based on metal nanozymes [74]. For a review of colorimetric sensing methods not discussed here, see the review by Aldewachi et al. [80].

### 5.1. Colorimetric Sensing Based on PMNP Formation and Growth

Synthesis of silver and gold PMNPs commonly involves the reduction in metal salts, e.g., Ag+ and [AuCl_4_]^−^, to form PMNPs [79,81]. Many sensing methods based on formation and/or growth of PMNPs take advantage of in situ formation of PMNPs causing significant visible colour changes when a target analyte is present [82]. In some cases, the target analyte serves as a direct reducing agent, e.g., for the colorimetric hydrazine sensor reported by Khan et al. [83], where upon the interaction of the dispersion with hydrazine, Ag^+^ ions are reduced forming AgNPs and producing a distinct colour change. In other cases, the target analyte acts to initiate the reduction of silver or gold by a reagent contained in the assay. This method can be seen in the work by Sivakumar, Park and Lee [79], where the process of colorimetric detection of pathogens by formation of AgNPs can be seen in Figure 9. This formation-based colorimetric detection functions by using the reducing agent quercetin added to the assay. In the presence of pathogen DNA, it forms complexes with Ag^+^ ions present in the assay [79]. Quercetin then acts to reduce the Ag^+^ ions to form AgNPs. The formation of AgNPs provides a notable change from colourless to red/brown in the presence of pathogens, where intensity of the colour is directly proportional to the concentration of pathogen DNA [79]. This mechanism allows for a reasonably sensitive, easy to use, naked eye detection using PMNPs. Although many colorimetric sensing methods based on PMNP formation and growth rely on detection by presence of formation, some others rely on the lack thereof; for example, Su et al. [84] recently prepared a novel colorimetric sensor based on the inhibition of photoinduced AuNP formation for the detection of 2-mercaptobenzothiazole (MBT).

### 5.2. Colorimetric Sensing Based on PMNP Aggregation

Aggregation of PMNPs feature several characteristics that can be utilized by colorimetric detection methods. Upon PMNP aggregation, a significant shift in the LSPR peak occurs due to LSPR interactions and overlaps that manifest in visible colour changes [85,86]. Aggregation of AuNPs and AgNPs can be generally induced by a change in salt concentration, pH or temperature [77,85,87]. Silver and gold nanoparticle aggregation can also be caused by specific molecules such as charged species, proteins, etc. [72]. Given this, it is difficult to apply aggregation-based colorimetric sensing techniques to diverse real-world samples due to unspecific aggregation, limiting sensitivity and selectivity [72]. However, Song et al. [72] prepared a 3-D colorimetric sensor platform, in order to overcome these challenges, called a bead-based system (BBS). AuNPs suspended in a BBS enable improvements in sensitivity and selectivity in complex matrices by maintaining NP mobility and stability. Song et al. [72] tested the reactivity of BBS as a colorimetric sensor by adding 1,4-dithioreitol (DTT) to induce aggregation in samples of AuNPs in BBS prepared under varying voltages (0–5.0 kV). The authors found that the samples containing smaller BBS, which were prepared at a higher voltage, reacted faster than those prepared at a lower voltage (larger size) [72]. The colour shift taking place during the reaction of the BBS is caused by the target molecule, DTT, diffusing into the BBS and reacting with AuNPs, leading to aggregation and consequently visible colour changes [72]. The stability of the BBS was also tested in buffer and various complex human and environmental samples and compared to a typical solution-based system (SBS) [72]. The BBS outperformed the SBS, where it showed the original properties of the AuNPs after 50 days in solution [72]. Zhang et al. [88] recently prepared a colorimetric sensing platform for detection of hydrochloric acid by redispersion of AuNP aggregates. The authors utilized glutathione-modified AuNPs that could be induced to aggregate by the amino/carboxyl-binding effect, electrostatic effect and the centrifugal effect, before interaction with HCl [88]. When introduced to HCl, the rapid redispersion of aggregated particles occurred due to protonation of the amino and carboxyl groups of glutathione in acidic environments and re-stabilization of electrostatic repulsions between NPs [88]. Therefore, presence of HCl in this sensing platform could be detected visually by fading of colour or by spectroscopic methods with the dampening of the LSPR peak of AuNPs [88].

### 5.3. Colorimetric Sensing with PMNP-Based Nanozymes

PMNP nanozymes have been of recent interest in biosensing applications due to their enzyme-like characteristics coupled with their intrinsic plasmonic behaviour [74]. Nanozymes can often exhibit superior properties compared to natural enzymes due to tunable catalytic activity, simple synthesis and ease of storage. As well, natural enzymes are more expensive, less stable and cannot be catalytically tuned or recycled [74]. Kumar et al. [74] prepared an array-based colorimetric sensor using copper, nickel and cobalt NPs to decorate carbon nanotubes in the presence of a cationic receptor to identify eight different pesticides. This nanozyme sensor array shows the positive detection of various pesticides in a colour pattern shown in Figure 10A [74]. Authors reported that their nanozyme-based sensor array can distinguish pesticides at concentrations as low as 10 μM and detect pesticides in a range as low as 1 to 8 μM [74]. Fu et al. [89] also recently utilized gold nanozymes to perform a colorimetric detection of iodide and indirectly mercuric ions (Hg^2+^) in the presence of 3,3′,5,5′-tetramethylbenzidine (TMB). Taking advantage of histidine-stabilized gold nanoclusters (His-AuNCs) with peroxidase-like activity, interactions with iodide led to colorimetric signalling by altering catalytic properties of gold nanoclusters and inducing aggregation [89]. This sensing method provides a simple and convenient means of detecting I^−^ and Hg^2+^ down to nanomolar concentrations by visual or UV-Vis detection [89]. Other works pertaining to colorimetric sensing with PMNP-based nanozymes include those by Duan et al. [90] who prepared a trimetallic AgPt-Fe_3_O_4_ nanozyme sensing method for CO (carbon monoxide) detection and Sang et al. [91] who utilized bimetallic CuAg nanoflowers with nanozyme properties for the colorimetric detection of acid phosphatase.

### 5.4. Colorimetric Sensing with Functionalized PMNPs

Gold and silver NPs allow for ease of surface functionalization, leading to increased selectivity, specificity and molecular tunability for sensing of a diverse array of targets, not only applicable in biosensing, but also drug-delivery, photoelectrical systems and response-triggered nanomaterials [92]. Behera et al. [78] took advantage of these properties to prepare a lateral flow biosensor to detect polyethylene terephthalate (PET), to overcome previous challenges of microplastic detection including extensive preprocessing and complex instrumentation (Figure 10B). The authors functionalized the surface of AuNPs with a synthetic peptide (SP1), which can bind PET [78]. AuNP-linked SP1 peptides are located in the sample pad (left side of device) before use, and when the sample is added, it flows towards the absorption pad (right side of device), where a simple schematic of a lateral flow biosensor can be seen in Figure 10B [78]. As the sample flows from the sample pad to the absorption pad (left to right), it passes both the test line and control line [78]. The test line only binds AuNP linked SP1 peptides bound to PET microplastic, and the control line binds AuNP linked SP1 peptides not bound to PET microplastic [78]. When the functionalized AuNPs are stopped at either of these points, aggregation occurs giving rise to a red colour which allows for visible colorimetric detection of positive or negative tests [78]. Similar applications to that of AuNPs in lateral flow sensing has been demonstrated by Behera et al. [78] and Liu et al. [93] who reported detection of flutriafol in food products. Bahamondes Lorca et al. [94] also prepared a lateral flow system for colorimetric sensing, using spherical copper NPs with a gold shell (Cu@AuNPs). It has been confirmed that the Cu@AuNPs could be functionalized with various antibodies, allowing for diverse biological detection [94]. In addition to the confirmation of broad functionalization ability, the authors stated that signal efficiency and specificity is similar to that of pure AuNPs, making colorimetric sensing applications of Cu@AuNPs a feasible cost-effective alternative [94].

## 6. LSPR-Based Sensing with PMNPs

This section overviews methods that monitor/trace LSPR peaks of PNMPs including PMNP-relevant SPR, SLR (surface lattice resonance sensing) and circular dichroism sensing.

Kim et al. [95] reported an approach to overcome the false negatives common for lateral flow assays and the complex DNA extraction step of PCR tests by preparing an assay which utilizes LSPR for detection of COVID-19. Authors integrated recombinant angiotensin-converting enzyme-2 (ACE2), acting as the receptor for SARS-CoV and SARS-CoV-2, into liposomes that were linked to AuNPs forming sensor arrays on glass [95]. Schematic of clinical sample acquisition and the mechanism of virus detection is shown in Figure 11A, respectively. Detection of the SARS-CoV-2 spike proteins have been identified through LSPR shifts between 5 and 25 nm upon binding of the spike protein (Figure 11B) [95]. Overall, this method of detection offers a low detection limit of 10 pg/mL and high ease of use, making it promising for early and simple clinical diagnosis [95].

Behrouzi and Lin [96] recently developed a sensing platform for the detection of SARS-CoV-2 nucleocapsid proteins utilizing PMNPs for LSPR detection. Aggregation of PMNPs occurs when in contact with the nucleocapsid proteins due to functionalization with NHS-esters to bind antibodies [96]. Detection is performed by either the naked eye or monitoring LSPR shifts by UV-Vis spectroscopy [96]. To prepare samples, droplets of viral solution and antibody-coated AuNPs are mixed [91,92,93,94,95,96]. When target nucleocapsid proteins are present, aggregation of AuNPs takes place shifting the LSPR peak [96]. Aggregation of PMNPs also gives rise to visible colour changes, where in this case the control sample is red and the sample positively detecting SARS-CoV2 nucleocapsid proteins is blue [96].

Kim et al. [46] has taken advantage of advanced PMNP morphologies to create a grating coupled SPR-CD (surface plasmon resonance-circular dichroism) sensor. Kim et al. [46] fabricated these sensors by transferring helicoid AuNPs onto a gold film using an elastic PDMS mould (Figure 12A) and confirmed sample transfer success by scanning electron microscopy (Figure 12B) [46]. Circular dichroism as a novel sensing modality provides an advantage of reducing signal fluctuations and thus lowering the limit of detection of the sensor system by a factor of 50 compared to typical SPR testing [46]. This sensor achieved a sensitivity of 379.2 nm/RIU and a detection limit in the low mM range for D-glucose, paving the way for new sensitive and reliable methods of CD-SPR detection. Kim et al. [46] has taken advantage of advanced PMNP morphologies to create a grating coupled SPR-CD (surface plasmon resonance-circular dichroism) sensor. Kim et al. [46] fabricated these sensors by transferring helicoid AuNPs onto a gold film using an elastic PDMS mould (Figure 12A) and confirmed sample transfer success by scanning electron microscopy (Figure 12B) [46]. Circular dichroism as a novel sensing modality provides an advantage of reducing signal fluctuations and thus lowering the limit of detection of the sensor system by a factor of 50 compared to typical SPR testing [46]. This sensor achieved a sensitivity of 379.2 nm/RIU and a detection limit in the low mM range for D-glucose, paving the way for new sensitive and reliable methods of CD-SPR detection. Lv et al. [47] also take advantage of this morphology with their work on gold helicoid nanoparticles with palladium shells (Au@PdNPs) for plasmonic hydrogen sensing. Use of the helicoid gold core allows for improved plasmonic properties, while maintaining the desired exterior sensitivity to hydrogen of the palladium shell [47].

Li et al. [97] recently developed a novel nanozyme-linked immunosorbent SPR biosensor for detection of cancer biomarkers. Integrating SPR into ELISA enabled more sensitive detection of biomarkers compared to visual observation of colour changes or UV-Vis detection (Figure 13A,B). Authors utilized Au@Pt nanozymes that mimic catalytic activity of natural enzymes, overcoming inherent instability of natural enzymes and having an advantage of ease of surface modification and lower production costs [97]. This highly sensitive detection method enabled a rapid and easy-to-use platform for biosensing in cancer screening and early diagnostics [97].

Sensing platform based on non-closely packed (ncp) AuNP arrays fabricated through a self-confined solid-state dewetting mechanism has been reported by Chen et al. (Figure 13C,D) [98]. In this process, thin gold film is deposited onto a substrate, such as quartz, and thermally dewetted to form ordered arrays of NPs without relying on costly nanofabrication [98]. The resulting ncp AuNP arrays exhibit strong surface lattice resonance (SLR) effects when excited with normal white-light incidence, which significantly enhances their optical response compared to traditional LSPR [98]. For sensing applications, the nanoparticle surfaces are functionalized using EDC/NHS chemistry to immobilize biomolecules such as protein A, enabling selective binding of IgG antibodies [98]. As molecules bind to the nanoparticle surfaces, they cause measurable shifts in the SLR peak position, which can be monitored in real time using a simple transmission setup: white light passes perpendicularly through the chip integrated in a PDMS microfluidic channel, and the transmitted light is collected by a spectrometer [98]. This method enables sensitive, reproducible and portable plasmonic biosensing without the need for bulky Kretschmann-based configurations.

Lin et al. [99] designed an SPR biosensing platform designed for multiplex detection of bladder cancer-associated miRNAs. Panel I of Figure 14 shows the recognition step, where two engineered TDNs—TDNsA and TDNsB—are programmed to selectively hybridize with miR-183 and miR-155, respectively [99]. These nanoswitches incorporate biotinylated reporter units (RA and RB) that are tethered to triplex DNA structures, allowing pH-controlled conformational switching and release [99]. Figure 14 illustrates the working principle of the triplex DNA nanoswitch (TDNs) with Panel II of Figure 14 depicting the sensing workflow on an SPR chip functionalized with S9.6 antibodies that bind DNA/RNA duplexes. Following the binding of TDNs/miRNA complexes to the surface, streptavidin-coated AuNPs (Strep-AuNPs) are introduced, which couple to the biotin-labelled reporter units, thereby amplifying the plasmonic response [99]. The multiplexing capability is achieved by sequential pH modulation: at pH 5.0, the C-G·C+ triplex structure of SA reconfigures, releasing AuNP-labelled RA and producing a decrease in the SPR signal proportional to the amount of miR-183 captured [99]. At pH 8.3, the T-A·T triplex of SB dissociates, releasing AuNP-labelled RB and causing a further signal decrease corresponding to miR-155 levels [99]. Panel III of Figure 14 shows the resulting real-time SPR sensorgram, where discrete shifts in reflectivity (ΔRU) correlate with the sequential release of AuNP reporters at defined pH values, enabling quantitative and selective detection of multiple miRNAs in a single sensing channel [97,98,99]. This strategy successfully integrates the programmability of pH-responsive DNA triplexes with nanoparticle-amplified SPR, enabling label-free, amplification-free and multiplex biosensing [99].

## 7. Directions of Future Developments and Advances in Sensing with PMNPs

The field of sensing with PMNPs has advanced significantly over the last 3–4 years with new highly evolved PMNP morphologies, advances in sensor fabrication and notable progress in both colorimetric and LSPR-based sensing that is translating into more advanced point-of-care sensing prototypes. PMNPs have found important applications in medical diagnostics and biochemical detection and monitoring. Contributing to the future of medical diagnostics, PMNPs have been implemented into the sensing of prevalent viruses and diseases, such as COVID-19 and cancer, providing ease of use and early detection capabilities. In biochemical measurements and monitoring, PMNPs have provided simplified detection methods for toxins, metals, plastics and more, applicable to real world samples. The inherent sensitivity of PMNPs due to their plasmonic properties has enabled very sensitive detection, while the modularity of PMNP properties (i.e., size, shape and surface functionalization) has allowed for selectivity to be customized to suit specific requirements. This review has shown the high value of PMNPs and their LSPR properties for their use in colorimetric and plasmonic-based sensing.

Along the lines of the recent progress summarized in this review, we reflect on the following potential directions of future perspective advances in the field in this concluding part:

(1) With significant progress made in the understanding of size and shape selection of PMNPs, development of scalable and reproducible synthetic procedures for industrial applications can still benefit from more work on scalability and reproducibility. This is especially true for anisotropic PMNPs that offer both better LSPR tunability and higher sensing sensitivity.

(2) Transition from PMNP dispersions to sensing substrates could take advantage of more standardization in order to universally utilize PMNPs made through different synthetic procedures, e.g., with different ligands and varying surface chemistries. That is one of the most important steps in fully translating PMNP properties into sensing designs.

(3) Formation of the ordered PMNP arrays to control interparticle distance is another important direction that is closely related to 2). The ordered PMNP arrays can offer both higher sensitivity and better reproducibility of sensing. One approach to achieve the interparticle distance control is through the encapsulation in a dielectric shell that can be subsequently removed. Bottlenecks in this approach include challenges in formation of heterointerfaces, as well as simultaneous preservation of colloidal stability and structural integrity of the shells. For instance, silica is known to impart excellent colloidal stability, and it is thus commonly used for dielectric shells on PMNPs. At the same time, silica has an appreciable aqueous solubility that appreciably limits applications of thin silica encapsulating shells. An alternative approach to ordered arrays is to use patterned surfaces for either directed self-assembly or integrating PMNPs with nanopatterned metal gratings.

(4) Integrating multiplexing into existing plasmonic sensor designs through utilization of several types of anisotropic PMNPs with readily tunable and appropriately spaced out LSPR peaks is a promising new dimension that can enable simultaneous detection of multiple analytes to greatly expand current sensing capabilities.

All these future directions summarized above, especially when combined, are envisioned to empower the development of a universal sensing platform that can be readily adapted to detect a broad variety of analytes, especially in the point-of-care diagnostics. Sustainability and scalability will also shape the next generation of PMNP technologies. Developing greener synthetic routes, employing recyclable materials and ensuring biocompatibility will be essential for responsible commercialization and clinical translation. Ultimately, the evolution of PMNP-based sensing toward integrated, intelligent and environmentally conscious systems will redefine the accessibility and impact of chemical and biological detection technologies across healthcare, environmental monitoring and industrial analytics.

## Figures and Tables

**Figure 1 molecules-30-04745-f001:**
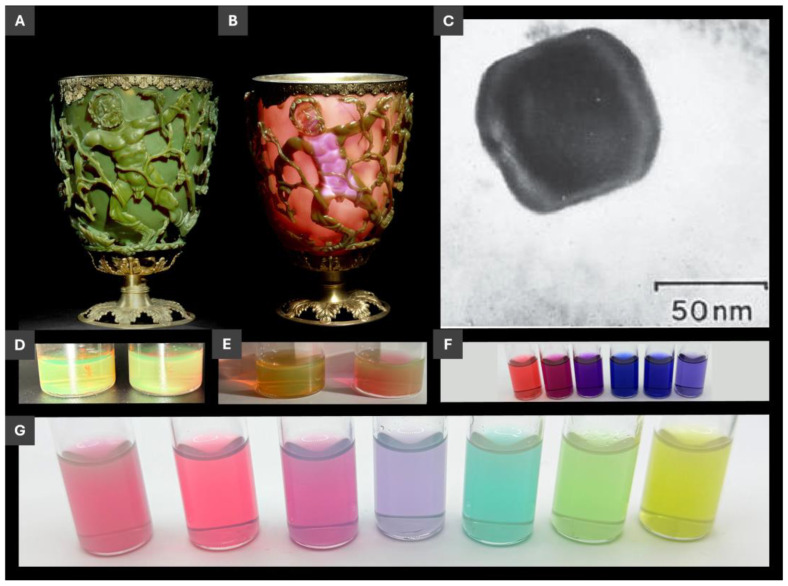
(**A,B**) Optical photographs of the Lycurgus cup (**A**) in reflected and (**B**) in transmitted light; (**C**) transmission electron microscopy (TEM) image of a silver-gold alloy PMNP within the glass of the Lycurgus cup; (**D**–**G**) optical images of PMNP dispersions from authors’ laboratory. (**A**–**C**) adapted with permissions from refs. [4,5]. **©** The Trustees of the British Museum. Shared under a Creative Commons Attribution-NonCommercial-ShareAlike 4.0 International (CC BY-NC-SA 4.0) licence. (**D**,**E**) PMNPs with dichroism similar to that if the Lycurgus cup described in refs. [6,7]. (**F**) PMNPs described in ref. [8]. and (**G**) PMNPs described in refs. [7,9].

**Figure 2 molecules-30-04745-f002:**
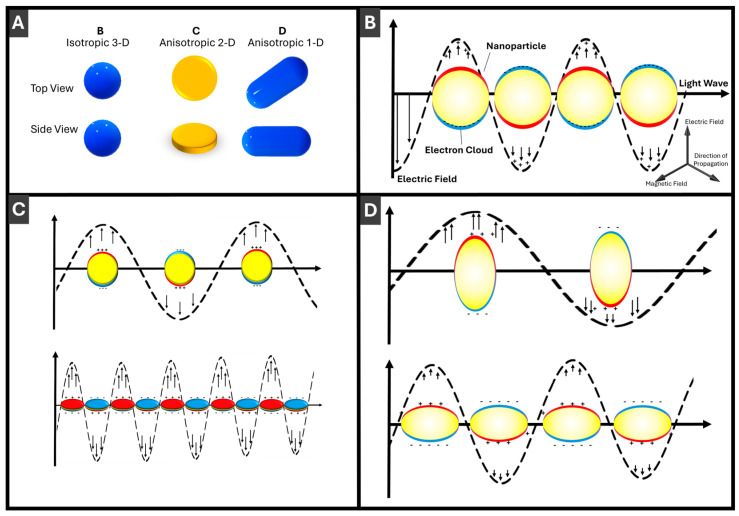
Schematic illustration of localized surface plasmon resonance (LSPR) in PMNPs: (**A**) comparison of three different PMNP morphologies: 3-D, 2-D and 1-D; (**B**) LSPR for 3-D (isotropic, spherical) PMNPs; (**C**) LSPR for anisotropic 2-D (disc-shaped) PMNPs; and (**D**) LSPR for anisotropic 1-D (rod-shaped) PMNPs. Direction of light propagation and components of the electromagnetic wave are shown in Panel (**B**). Incident light induces collective oscillations of the PMNP electron cloud relative to the positively charged lattice, generating a strong localized electromagnetic field indicated by arrows.

**Figure 3 molecules-30-04745-f003:**
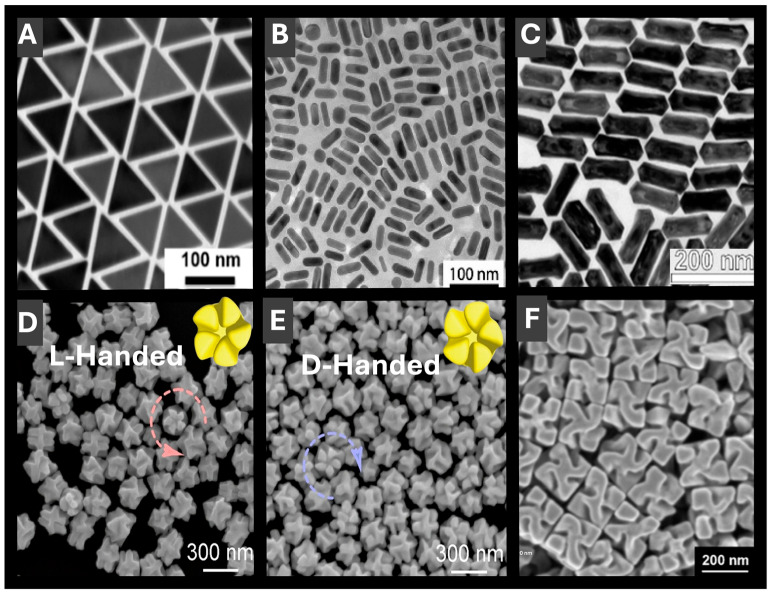
Electron microscopy (EM) images of representative anisotropic PMNP morphologies with tunable optical properties for LSPR sensing. (**A**) Two-dimensional anisotropic gold nanotriangles with a narrow size distribution. Reprinted with permission from [39]. Copyright 2022 American Chemical Society. (**B**) One-dimensional hollow gold-silver nanorods with enhanced chemical stability arising from gold-rich outer regions. Reproduced from ref. [42]. (**C**) Anisotropic gold-silver nanorattles with LSPR spanning from 1000 to 3000 nm. Reprinted with permission from [43]. Copyright 2024 American Chemical Society. (**D**,**E**) Gold nanorods with right- and left-handed five-fold rotational symmetries, advantageous for chiral LSPR sensing. Reproduced from ref. [44]. (**F**) Symmetric helicoid PMNPs with a high g-factor enabling sensitive chiral detection. Reproduced from ref. [46].

**Figure 4 molecules-30-04745-f004:**
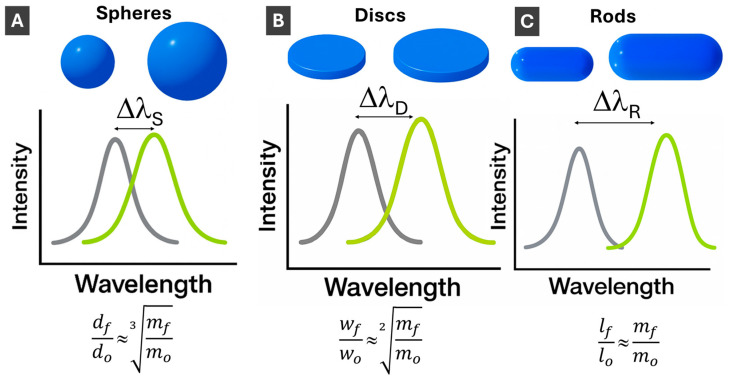
Schematics illustration of LSPR changes upon PMNP size increase for (**A**) 3-D spherical; (**B**) 2-D disc-shaped; and (**C**) 1-D rod-shaped PMNPs. **Δλ_Spheres_** < **Δλ_Discs_** < **Δλ_Rods_**. Scaling laws show how much material (mass final, ***m_f_***, relative to initial mass, ***m**_o_***) is needed to increase diameter (from ***d_o_*** to ***d_f_***) for 3-D PMNPs, width (from ***w_o_*** to ***w_f_***) for 2-D PMNPs and length (from ***l_o_*** to ***l_f_***) for 1-D PMNPs.

**Figure 5 molecules-30-04745-f005:**
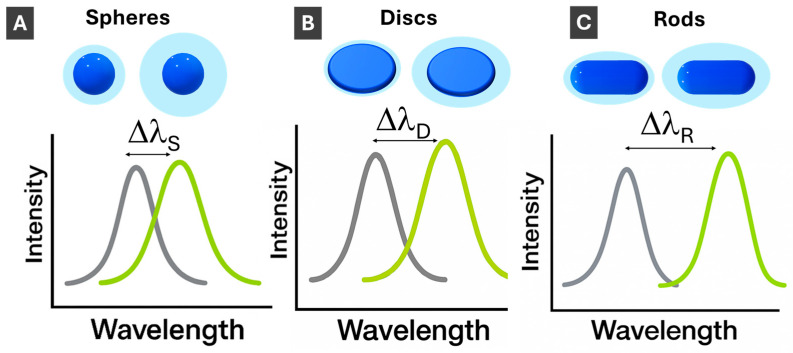
Schematics illustration of LSPR changes with increasing refractive index, ***n***, of a surrounding medium for (**A**) 3-D spherical; (**B**) 2-D disc-shaped; and (**C**) 1-D rod-shaped PMNPs. **Δλ_Spheres_** < **Δλ_Discs_** < **Δλ_Rods_**.

**Figure 6 molecules-30-04745-f006:**
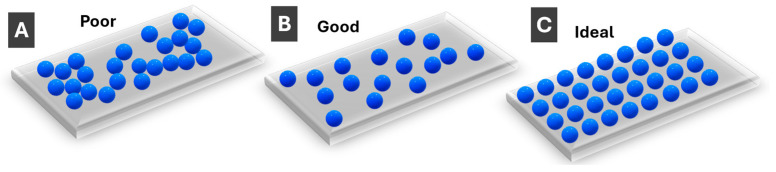
Schematic illustration of monolayers of (**A**) clustered/aggregated PMNPs (poor); (**B**) randomly well-spaced PMNPs (good); and (**C**) regularly spaced PMNPs (ideal).

**Figure 7 molecules-30-04745-f007:**
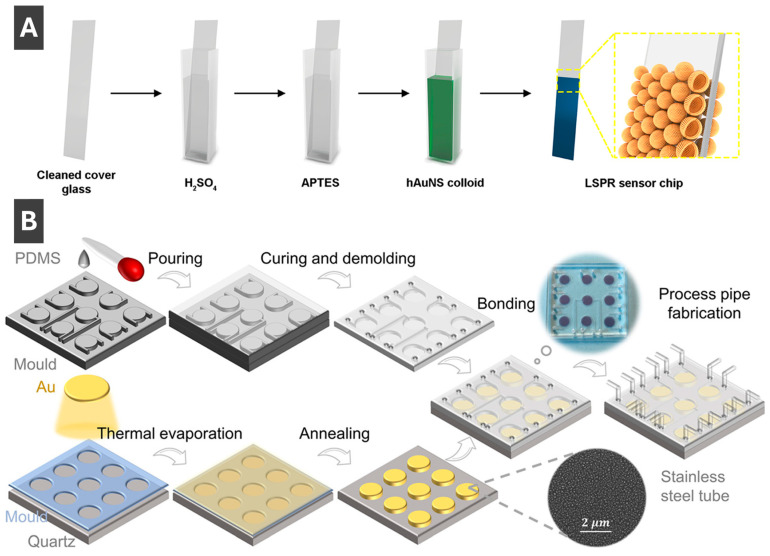
(**A**) Fabrication of LSPR sensor chip by immobilizing hollow gold nanoshells onto functionalized glass slide. Reprinted with permission from [69]. Copyright 2024 American Chemical Society. (**B**) Schematic fabrication process of a microfluid chip integrated with AuNPs for LSPR sensing of carcinoembryonic antigen in human serum. Reproduced from ref. Reprinted with permission from [70]. Copyright 2022 American Chemical Society.

**Figure 8 molecules-30-04745-f008:**
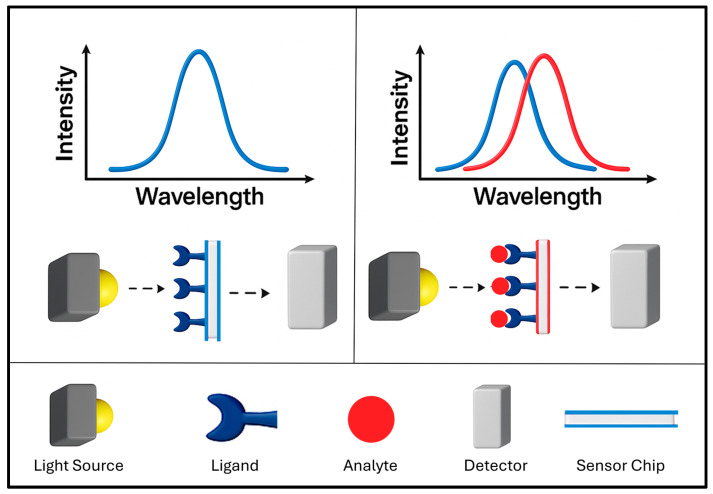
Schematic representation of LSPR-based colorimetric sensing with PMNPs. Changes in the local environment or target binding events produce measurable LSPR shifts. These shifts can also correspond to visible colour changes, illustrated here by the transition from an original LSPR peak (blue) to an LSPR peak (red) upon analyte binding.

**Figure 9 molecules-30-04745-f009:**
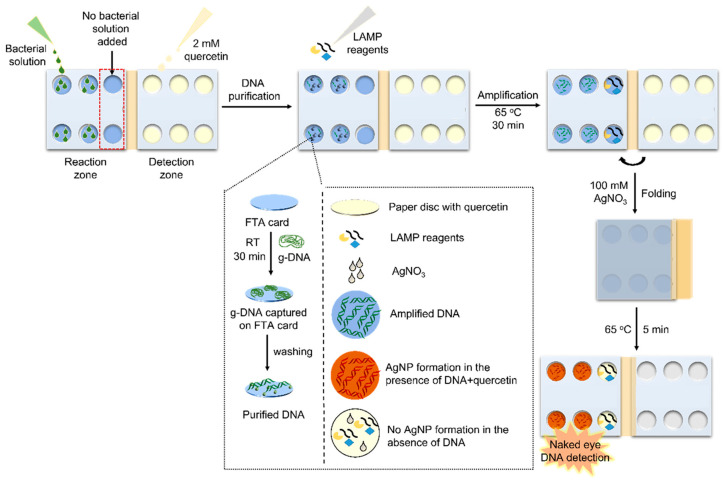
Schematic of naked eye colorimetric detection microdevice for detection of pathogens by the formation of AgNPs upon interaction with pathogen DNA and quercetin. Reproduced from ref. Reprinted with permission from [79]. Copyright 2023 American Chemical Society.

**Figure 10 molecules-30-04745-f010:**
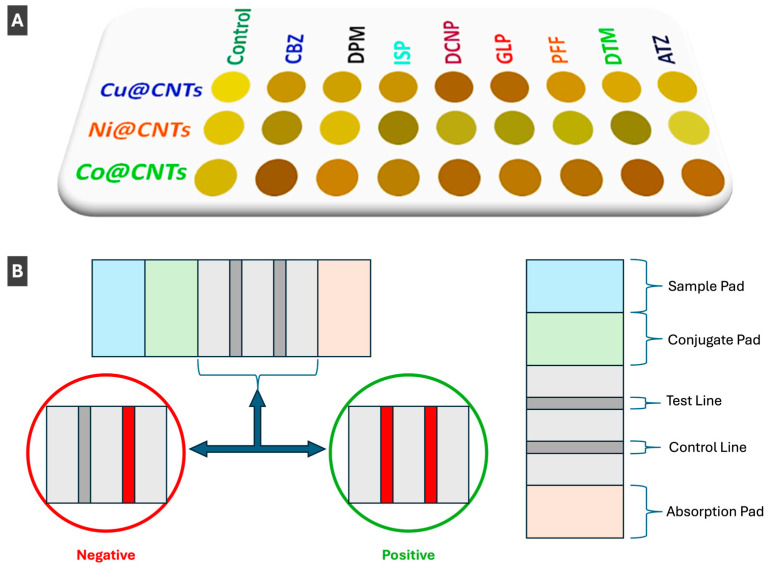
(**A**) Array-based colorimetric nanozyme sensor utilizing metal nanoparticle (Cu, Ni, Co) decorated carbon nanotubes (CNTs) for the pattern-based recognition and detection of eight common pesticides. Reprinted with permission from [74]. Copyright 2024 American Chemical Society. (**B**) Simplified schematic of lateral flow biosensor, showing optical outcomes of negative and positive detection, described in ref. [78].

**Figure 11 molecules-30-04745-f011:**
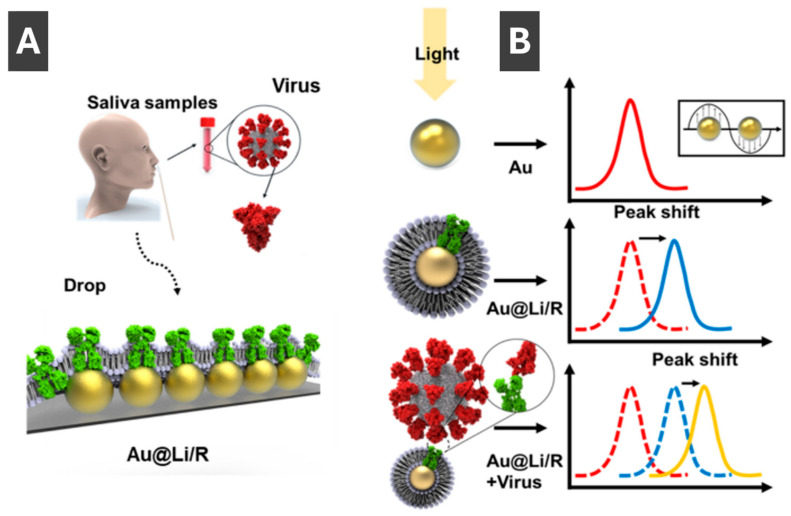
(**A**) Schematic image of clinical sample acquisition onto recombinant protein-embedded liposome functionalized AuNP (Au@LiR) LSPR sensor. (**B**) Mechanism of virus detection with LSPR peak shifts between AuNPs, Au@Li/Rs and Au@Li/R upon virus binding. Reproduced with permission from ref. [95].

**Figure 12 molecules-30-04745-f012:**
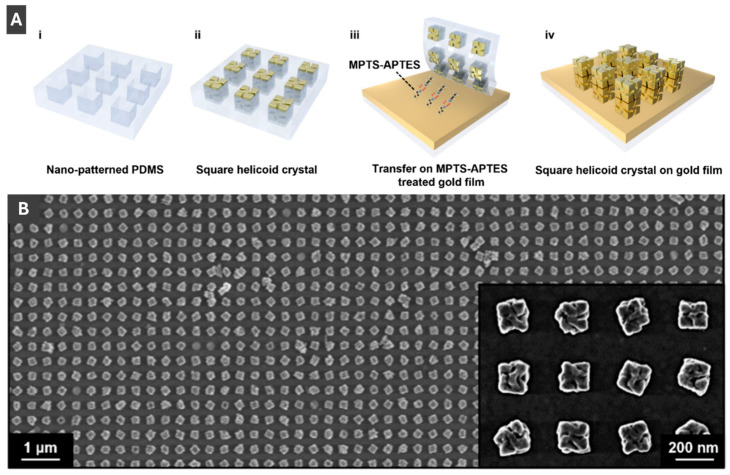
(**A**) Schematic of fabrication process of square helicoid crystal on gold film using MPTS-APTES. (**B**) Scanning electron microscopy image of 180 nm helicoids uniformly transferred onto gold film. Reprinted with permission from [46]. Copyright 2024 American Chemical Society.

**Figure 13 molecules-30-04745-f013:**
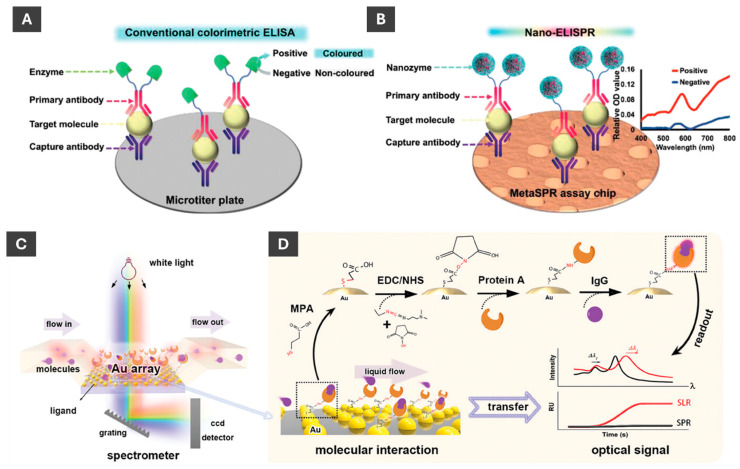
Schematic comparison of sensing by ELISA and nano-ELISPR biosensors. (**A**) Typical ELISA detection platform process with detection via colour, i.e., optical density (OD) changes. (**B**) Nano-ELISPR biosensor detection platform process with detection based on spectral changes. (**A,B**) These are reproduced with permission from ref. [97]. (**C**) Fundamental mechanism of surface lattice resonance and SPR measurement, which relies on ncp AuNP array to detect molecular interactions. (**D**) Schematic of surface functionalization of AuNPs with (EDC/NHS) and linking protein A so that a measurable optical signal upon molecular interaction between protein A and IgG can be realized. (**C**,**D**) are reproduced with permission from ref. [98].

**Figure 14 molecules-30-04745-f014:**
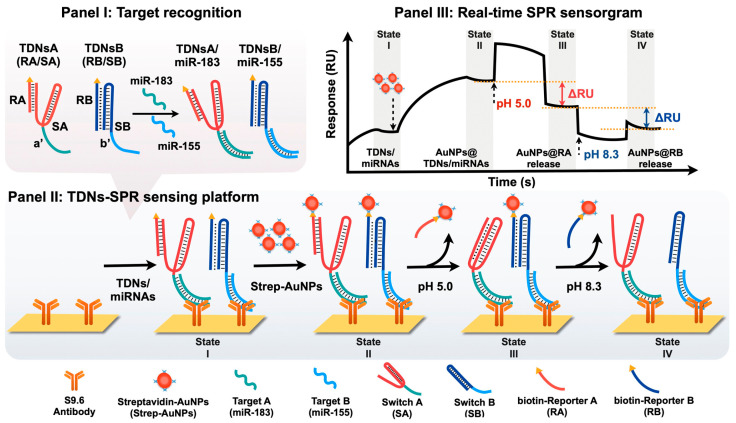
Schematic diagram and SPR measurement of pH-responsive triplex DNA nanoswitches. (**Panel I**) Specific microRNA sequences to be incorporated into the SPR platform. (**Panel II**) AuNPs functioning as amplifying labels for SPR detection of specific microRNAs, where pH changes trigger release of streptavidin conjugated AuNPs. (**Panel III**) SPR response curve for this sensing process. Reprinted with permission from ref. [99]. Copyright 2025 American Chemical Society.

## Data Availability

No new data were created or analyzed in this study.

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
