# Peer review of "Plasmonic Metal Nanoparticles in Sensing Applications: From Synthesis to Implementations in Biochemical and Medical Diagnostics"

_molecules, 2025, doi:10.3390/molecules30244745_

Round 1
Reviewer 1 Report
Comments and Suggestions for Authors
Manuscript by Nemeth et al. presents a review of recent progress on selected aspects of using plasmonic nanoparticles in biosensing. Using of noble metals nanoparticles is progressing in two recent decades, with an incredible development on the fields of synthesis, chemical functionalizations, and applications in sensing and optical measurements. Thereofore, it is a hard task to review the progress in last few years without any specialization of the field of interest from this whole topic. This is the case of this work, where Authors decided to focus on the selected aspects of synthesis of PMNPs and thier utilization in sensing/biosensing, colorometic and LPSR sensing. I apreciate the broad literature composed by very recent papers from these topics. However, there are some points which needs some improvements, clarifications or corrections.
- The structure of the manuscript is not uniform. It starts with very basic introduction, after which there appear some paragraphs which are rather "review of reviews", where each sentence directs to another review, without any deeper discussion. Desipte I understand that it is aimed to direct the reader to other specialized reviews, however then it is to short and not structurized enough. I would suggest to support this part with some table (for example in SI) with the topics (synthesis, chemical modifications, etc.) and selected references to other reviews. Next, there are chapters about the synthesis, which are not so detailed in some parts, and then there are well-balanced chapters about sensing applications.
- About the introduction - this is in my opinion the weakest part of this manuscript. It starts with rather standard story about Lycurgus cup and the tunability of the PMNPs solutions when it comes to colour, which then is connected with the basics of the plasmon resonance. This suggests that the idea is to provide solid foundations for the discussion of the mechanisms discussed in the further part. However, there are essential lacks and inconsistencies. Authors do not define clear criteria to assign the structures as 1D, 2D, or 3D, and are not consequent with this description (e.g. nanorods once are discussed as 1D material, but are named as 2D elsewhere). There is mentioned that 3D is for isotropic structures, shouldn't it be rather 1D then, if only one dimension (diameter) can be tuned?
- There is mentioned LSPR and shifting of the resonance peak, but we do not see any corresponding spectrum - it would be extremely beneficial to show correspondence between not only size of PNMP and colour or the light wave, but with the spectrum, how it shifts with changing size and the shape. Now this discussion is a bit chaotic. Moreover, it would be beneficial to include here also information about the role of the environment on the position of plasmon resonance, as it is important effect in the presented results.
- Figure 2 - it is not clear what is "light wave" as shown in the horizontal axes, is it in space, or in time? Is this just the direction of propagation? Why in panel A there is much more arrows inside the wave, is there any reason for that? Similarly, why part of the panel C is drawn with different style?
- When nanorattles are discussed (2.1, Fig. 3D-F), there is no information about the synthesis of such structures, however, this paragraph in devoted to the synthesis and postsymthesis procedures used to obtain given structures.
- Page 12, line 421 - when Fig. 9 is mentioned in the text, it should be mentioned that it is the Fig. 9A and B that is discussed at this point.
- Line 27 - good metals are mentioned, it should be rather noble or precious metals.
Author Response
Response to the reviewers’ comments
Reviewer 1:
Manuscript by Nemeth et al. presents a review of recent progress on selected aspects of using plasmonic nanoparticles in biosensing. Using of noble metals nanoparticles is progressing in two recent decades, with an incredible development on the fields of synthesis, chemical functionalizations, and applications in sensing and optical measurements. Thereofore, it is a hard task to review the progress in last few years without any specialization of the field of interest from this whole topic. This is the case of this work, where Authors decided to focus on the selected aspects of synthesis of PMNPs and thier utilization in sensing/biosensing, colorimetric and LPSR sensing. I apreciate the broad literature composed by very recent papers from these topics. However, there are some points which needs some improvements, clarifications or corrections.
Thank you for the nice summary of our work, positive comments, and constructive critical suggestions that we have addressed below.
1) The structure of the manuscript is not uniform. It starts with very basic introduction, after which there appear some paragraphs which are rather "review of reviews", where each sentence directs to another review, without any deeper discussion. Desipte I understand that it is aimed to direct the reader to other specialized reviews, however then it is to short and not structurized enough. I would suggest to support this part with some table (for example in SI) with the topics (synthesis, chemical modifications, etc.) and selected references to other reviews. Next, there are chapters about the synthesis, which are not so detailed in some parts, and then there are well-balanced chapters about sensing applications.
Based on your instructive suggestions, the Table summarizing recent reviews is prepared, and an SI file is created to present this Table in the revised version of the manuscript.
2) About the introduction - this is in my opinion the weakest part of this manuscript. It starts with rather standard story about Lycurgus cup and the tunability of the PMNPs solutions when it comes to colour, which then is connected with the basics of the plasmon resonance. This suggests that the idea is to provide solid foundations for the discussion of the mechanisms discussed in the further part. However, there are essential lacks and inconsistencies. Authors do not define clear criteria to assign the structures as 1D, 2D, or 3D, and are not consequent with this description (e.g. nanorods once are discussed as 1D material, but are named as 2D elsewhere). There is mentioned that 3D is for isotropic structures, shouldn't it be rather 1D then, if only one dimension (diameter) can be tuned?
It was our plan to write a very brief introduction, since the general knowledge, e.g. on LSPR shifts, is well discussed in multiple excellent previous reviews (and we have provided the references). Figure 1 was made to appreciate LSPR in a simple and effective way - by different colours, as well as unique dichroic effects in strongly scattering PMNPs. Figure 2 provides a simple idea of LSPR resonances, while emphasizing the comparison between anisotropic and isotropic PMNPs that we feel is an important point to highlight. With respect to our description of anisotropic particles, we have now clarified Figure 2.
In “3-D” for morphological description of the growing shape/crystal, “3” refers to 3 orthogonal directions of growth available, e.g. for spheres, cubes, octahedra, etc. For 1-D morphologies, only a single dimension is available for growth resulting in rods, wires, nanotubes, etc. Correspondingly, 2-D structures are the structure grown in two dimensions of a plane, e.g. platelets, thin prisms, graphene, etc. At the same time, (definitely adding more complexity, so your questions are very understandable), relative to bulk, small NPs are often considered to be “0-D”, as per common description of quantum confinement in quantum dots.
Inspired by your comments, we have extended the corresponding description in the introduction and further expanded upon it in the new section on morphology-property relationship of PMNPs adding overall four new figures (Figs. 4, 5, 6, and 8). So, we hope that we fully address your points.
3) There is mentioned LSPR and shifting of the resonance peak, but we do not see any corresponding spectrum - it would be extremely beneficial to show correspondence between not only size of PNMP and colour or the light wave, but with the spectrum, how it shifts with changing size and the shape. Now this discussion is a bit chaotic. Moreover, it would be beneficial to include here also information about the role of the environment on the position of plasmon resonance, as it is important effect in the presented results.
We think that this information can be found in several excellent previous reviews in this field, that are cited in our review. At the same time, to address your comments, we have added Figure 4 to outline schematically LSPR changes upon changes in size for different morphologies, as well as Figure 5 showing LSPR changes upon changing the dielectric constant of surrounding medium.
4) Figure 2 - it is not clear what is "light wave" as shown in the horizontal axes, is it in space, or in time? Is this just the direction of propagation? Why in panel A there is much more arrows inside the wave, is there any reason for that? Similarly, why part of the panel C is drawn with different style?
The higher density of arrows is meant to represent higher energies of transverse resonances. Based on your comments, we have revised this figure showing directions of propagation and more uniform arrow density for simplicity, including several revisions of the panels making more systematic presentation.
5) When nanorattles are discussed (2.1, Fig. 3D-F), there is no information about the synthesis of such structures, however, this paragraph in devoted to the synthesis and postsymthesis procedures used to obtain given structures.
Based on your instructive comments, we have extended the description on nanorattles in the revised manuscript, elucidating their formation for the readers.
6) Page 12, line 421 - when Fig. 9 is mentioned in the text, it should be mentioned that it is the Fig. 9A and B that is discussed at this point.
Thank you for pointing this out! In the revised manuscript, we have properly referred to an older Fig. 9A,B (now Fig. 13A,B). As well, we have double-checked other figures and added a reference to older Fig. 6B.
7) Line 27 - good metals are mentioned, it should be rather noble or precious metals.
That is a good point to consider/discuss. Good metals refer to the metal that are close to the simple free-electron model or a subsequently introduced jellium model.
Upon a literature search, this term is not commonly used in recent literature on plasmonic NPs, compared to nanoclusters, so we agree to change it and did it in the revised manuscript.
“Free or delocalized electrons of metals are essential for PMNPs, as well as chemical stability to resist PMNP degradation. Silver and gold are two metals that meet these two criteria and thus are dominant materials of PMNPs.”
Reviewer 2 Report
Comments and Suggestions for Authors
This manuscript systematically reviews the research progress of plasma metal nanoparticles in the field of sensing in the past 3-4 years, covering core contents such as synthesis and modification, substrate preparation, and colorimetric/local surface plasmon resonance sensing applications. The topic selection has significant academic value and application prospects. However, the manuscript has problems such as the need to optimize its structural logic, insufficient depth in some parts, weak data support, and non-standard expression, which require major revisions.
- Although the current chapters cover the core content, the connection between each part is not tight enough, especially the connection between “PMNPs synthesis and modification” and “Sensing applications” is not clearly sorted out. It is suggested to add a transitional chapter to clarify the structure-activity relationship between the structural characteristics and sensing performance of PMNPs with different morphologies/functionalized features.
- In the manuscript, multiple sensing cases involve SERS technology. Is there a quantitative correlation between the density of "electromagnetic hotspots" formed by PMNPs assemblies and the enhancement factor of SERS signals?
- This work overviews recent (last 3-4 years) advances in sensing based on localized surface plasmon resonance of plasmonic metal nanoparticles. As far as I know, plasma metal nanoparticles also have many applications in the field of photoelectric sensing. It is suggested that the author summarize them in this review.
- The manuscript has some problems such as lengthy sentences and inconsistent use of terms (for example, the alternating use of “PMNPs” and “plasma metal nanoparticles” lacks regularity).
- The latest research in some fields (2024-2025) has insufficient citations, and the citation formats of some literature are not standardized (such as not clearly marking the corresponding figures/tables for the citation positions). It is necessary to supplement high-impact literature from the past 1-2 years, unify the citation format of literature.
- What are the main bottlenecks in the large-scale production of existing preparation technologies (such as dielectric shell packaging and patterned surface self-assembly) mentioned in the future development direction of “construction of ordered PMNPs arrays”?
Author Response
Reviewer 2:
This manuscript systematically reviews the research progress of plasma metal nanoparticles in the field of sensing in the past 3-4 years, covering core contents such as synthesis and modification, substrate preparation, and colorimetric/local surface plasmon resonance sensing applications. The topic selection has significant academic value and application prospects. However, the manuscript has problems such as the need to optimize its structural logic, insufficient depth in some parts, weak data support, and non-standard expression, which require major revisions.
We appreciate your time reviewing the manuscript and your insightful and instructive comments. We have introduced multiple changes to address them that are described below point-by-point.
1. Although the current chapters cover the core content, the connection between each part is not tight enough, especially the connection between “PMNPs synthesis and modification” and “Sensing applications” is not clearly sorted out. It is suggested to add a transitional chapter to clarify the structure-activity relationship between the structural characteristics and sensing performance of PMNPs with different morphologies/functionalized features.
Based on your instructive suggestions, we have expanded upon the transition from the synthesis and modification of PMNPs, introducing a new section on structure-property relationships of PMNPs. For this section, we have added Figures 4 and 5 to provide a schematic description of LSPR changes with respect to dimensional and environmental changes. We have also made and added a schematic figure on monolayers (Fig. 6) and LSPR sensing (Fig. 8).
2. In the manuscript, multiple sensing cases involve SERS technology. Is there a quantitative correlation between the density of "electromagnetic hotspots" formed by PMNPs assemblies and the enhancement factor of SERS signals?
It is a very good question. We have encountered little consistent conclusive information upon SERS enhancement factor in PMNP assemblies. The main reason for this, in our understanding, is that while synthesis of size- and shape-uniform PMNP is challenging, it is even more challenging to prepare uniformly reproductible PMNP assemblies to systematically study enhancement factors. Even for the top-down structures, the required precision presents appreciable challenges. In our experience with PMNP substrates, when measuring enhancement factors for the large ensembles of PMNPs, the best enhancement approaching 1010 was the easiest to produce for PMNPs that cannot pack in regular arrays, e.g. featuring five-fold symmetries, such as D5h pentagonal bipyramids (decahedra) and their stellated derivatives.
3. This work overviews recent (last 3-4 years) advances in sensing based on localized surface plasmon resonance of plasmonic metal nanoparticles. As far as I know, plasma metal nanoparticles also have many applications in the field of photoelectric sensing. It is suggested that the author summarize them in this review.
True, there are several other applications of PMNPs, including photothermal applications, catalysis, etc. They are not a direct subject of this review. At the same time, in a broader context of a general introduction, we agree that a brief summary of other applications of PMNPs with the references can be instructive, and we included it in the revised manuscript.
4. The manuscript has some problems such as lengthy sentences and inconsistent use of terms (for example, the alternating use of “PMNPs” and “plasma metal nanoparticles” lacks regularity).
Thank you for pointing it out. We have revised “plasmonic metal nanoparticles” to PMNPs for consistency. Several times, we have used the term “plasmonic nanomaterials” that is meant to imply a broader context of both NPs and engineered (e.g. top down) structures.
5. The latest research in some fields (2024-2025) has insufficient citations, and the citation formats of some literature are not standardized (such as not clearly marking the corresponding figures/tables for the citation positions). It is necessary to supplement high-impact literature from the past 1-2 years, unify the citation format of literature.
That was exactly our starting point working on this review: most recent and highest-impact papers, then we expanded in time and scope. At your request, upon further search we have been able to find and include five most recent works of high impact that could be included and 17+ new references overall.
6. What are the main bottlenecks in the large-scale production of existing preparation technologies (such as dielectric shell packaging and patterned surface self-assembly) mentioned in the future development direction of “construction of ordered PMNPs arrays”?
Thank you for a good question. We have expanded upon these points in the concluding part of the revised manuscript.
Round 2
Reviewer 1 Report
Comments and Suggestions for Authors
All my concerns were properly addressed. After revision this manuscript is a way better structured and thus much more clear. Newly added discussion sections fill the lacks of the previous version of this paper.
One additional issue - in Fig. 5 cartoons of the different types of nanoparticles are switched between B and C panels.
Author Response
Comments: All my concerns were properly addressed. After revision this manuscript is a way better structured and thus much more clear. Newly added discussion sections fill the lacks of the previous version of this paper.
One additional issue - in Fig. 5 cartoons of the different types of nanoparticles are switched between B and C panels.
Author's reply:
We really appreciate all your constructive leasing to manuscripts improvements. Thank you for looking carefully again and pointing out Fig. 5 - we will correct in the revised version. In addition to this revision of Fig. 5, we have spaced out LSPR peaks in Fig. 4 and Fig. 5 for better clarity for the readers.
Reviewer 2 Report
Comments and Suggestions for Authors
We are satisfied with the revision, thanks for the hard work authors made.
Author Response
We are satisfied with the revision, thanks for the hard work authors made.
Reply:
Thank you very much for all your time and instructive comments that made the manuscript better.